# Improved Genetic Map Identified Major QTLs for Drought Tolerance- and Iron Deficiency Tolerance-Related Traits in Groundnut

**DOI:** 10.3390/genes12010037

**Published:** 2020-12-30

**Authors:** Manish K. Pandey, Sunil S. Gangurde, Vinay Sharma, Santosh K. Pattanashetti, Gopalakrishna K. Naidu, Issa Faye, Falalou Hamidou, Haile Desmae, Ndjido Ardo Kane, Mei Yuan, Vincent Vadez, Shyam N. Nigam, Rajeev K. Varshney

**Affiliations:** 1International Crops Research Institute for the Semi-Arid Tropics (ICRISAT), Hyderabad 502324, India; g.sunil@cgiar.org (S.S.G.); s.vinay@cgiar.org (V.S.); santosh.pattanashetti@gmail.com (S.K.P.); v.vadez@cgiar.org (V.V.); snnigam1947@gmail.com (S.N.N.); 2College of Agriculture, University of Agricultural Sciences, Dharwad (UAS-D), Vijayapur 580005, India; naidug@uasd.in; 3Centre Régional d’Excellence Sur Les Céréales Sèches et Cultures Associées, Institut Sénégalais de Recherches Agricoles (ISRA), Bambey BP 53, Senegal; issafaye2001@yahoo.fr (I.F.); ndjidokane@gmail.com (N.A.K.); 4Sahelian Center, International Crops Research Institute for the Semi-Arid Tropics (ICRISAT), Niamey 12404, Niger; h.falalou@cgiar.org; 5International Crops Research Institute for the Semi-Arid Tropics (ICRISAT), Bamako BP 320, Mali; h.desmae@cgiar.org; 6Key Laboratory of Peanut Biology and Genetic Improvement, Shandong Peanut Research Institute, Qingdao 266100, China; yuanbeauty@126.com; 7Institut de Recherche pour le Developement (IRD), Université de Montpellier, UMR DIADE, 34394 Montpellier, France

**Keywords:** abiotic stress, *Arachis hypogaea*, map density, SNP array, genetic map, genomics-assisted breeding, peanut

## Abstract

A deep understanding of the genetic control of drought tolerance and iron deficiency tolerance is essential to hasten the process of developing improved varieties with higher tolerance through genomics-assisted breeding. In this context, an improved genetic map with 1205 loci was developed spanning 2598.3 cM with an average 2.2 cM distance between loci in the recombinant inbred line (TAG 24 × ICGV 86031) population using high-density 58K single nucleotide polymorphism (SNP) “Axiom_*Arachis*” array. Quantitative trait locus (QTL) analysis was performed using extensive phenotyping data generated for 20 drought tolerance- and two iron deficiency tolerance-related traits from eight seasons (2004–2015) at two locations in India, one in Niger, and one in Senegal. The genome-wide QTL discovery analysis identified 19 major main-effect QTLs with 10.0–33.9% phenotypic variation explained (PVE) for drought tolerance- and iron deficiency tolerance- related traits. Major main-effect QTLs were detected for haulm weight (20.1% PVE), SCMR (soil plant analytical development (SPAD) chlorophyll meter reading, 22.4% PVE), and visual chlorosis rate (33.9% PVE). Several important candidate genes encoding glycosyl hydrolases; malate dehydrogenases; microtubule-associated proteins; and transcription factors such as MADS-box, basic helix-loop-helix (bHLH), NAM, ATAF, and CUC (NAC), and myeloblastosis (MYB) were identified underlying these QTL regions. The putative function of these genes indicated their possible involvement in plant growth, development of seed and pod, and photosynthesis under drought or iron deficiency conditions in groundnut. These genomic regions and candidate genes, after validation, may be useful to develop molecular markers for deploying genomics-assisted breeding for enhancing groundnut yield under drought stress and iron-deficient soil conditions.

## 1. Introduction

Groundnut (*Arachis hypogaea* L.; AABB; 2n = 4X = 40) is an important grain legume and oilseed crop that is mainly grown as a rainfed crop in the semi-arid regions in the world. It is the third-largest oilseed crop after soybean and rapeseed with a global production of 45.8 million tons from 28.5 million hectares of cultivated area worldwide [1]. Global food security is continuously being haunted by the ever-increasing population and drastic and uncertain climate changes leading to more water scarcity and soil health deterioration [2]. Reduced precipitation and changing rainfall patterns cause a frequent onset of drought and rising temperatures around the world [3,4]. As two-thirds of global groundnut production occurs in rainfed areas with unpredictable and insufficient precipitation [5], drought stress causes a significant decline in crop yields due to its adverse effects on plant growth, physiology, and reproduction of the crop [6]. Iron deficiency in soil is another concern, which is estimated to occur in 30–50% of cultivated soils globally [7]. In India, one-third of the cultivated area has calcareous soils that are deficient in iron, mostly distributed in the low rainfall areas, where groundnut is a major crop and suffers from iron deficiency (ID) or interveinal chlorosis leading to a significant decrease in pod yield (16–32%) [8]. In Sub-Saharan Africa (SSA) arable lands (over 50% of the world’s potential land for cultivation), iron is among the five predominant soil micronutrients being identified as important for crop productivity [9]. Acute iron deficiency can lead to plant death and even complete crop failure. In order to sustain high productivity and wider adaptation under these challenging growing conditions, the development of new groundnut varieties with improved genetics considering such major constraints are required for better adoption in farmer’s field [10].

Low heritability, large genotype × environment interactions, and the polygenic genetic nature of complex traits pose serious challenges for genetic improvement in conventional varietal development programs, especially for water-limited environments and iron-deficient soils. Modern technologies and breeding strategies have great potential to improve such complex traits and achieve sustainable yield under water-limited and iron-deficit conditions. Additionally, to enhance varietal replacement rate in farmers’ fields [11], new methods and technologies such as genomics tools, rapid generation advancements, and genomic selection can help in developing better groundnut varieties at a faster pace [12,13,14]. Recently, the 5G breeding approach has been suggested by Varshney et al. [15] for integrating modern genomics and genetic technologies with crop improvement programs.

In light of the above, the identification of genomic regions and markers for drought tolerance- and ID tolerance- related traits is crucial for performing effective early generation selection. In general, quantitative trait locus (QTL) mapping [16] and association mapping [17] are used for the identification of molecular marker(s) associated with the trait. In this context, some studies dealing with the development of molecular markers and genetic maps were conducted in the past for undertaking QTL analysis in groundnut [18,19,20]. In fact, some QTL mapping studies have also been reported for drought tolerance- and ID tolerance-related traits [16,21,22,23,24]. However, the majority of these studies failed to provide conclusive results due to a lack of sufficiently dense genetic map (≈56–347 loci) and reference genome required for gene discovery underlying QTL regions. During the last decade, modern genotyping platforms such as GBS (genotyping by sequencing), WGRS (whole-genome re-sequencing), ddRAD-seq (double digest restriction site-associated DNA), and single nucleotide polymorphism (SNP) array were successfully deployed for trait mapping and high-resolution mapping in groundnut including the most recent deployment of SNP array in genomic selection [12]. High-density SNP array is a promising approach that generates high-quality genotyping data with minimum missing call rates and uniform genome coverage for high-resolution trait mapping. In groundnut, a 58K SNP “Axiom_*Arachis*” array is available and represents an important genotyping platform for performing high-resolution mapping [25]. With the availability of draft genome sequences of tetraploid cultivated groundnut [26,27,28], the high-density genotyping and sequencing-based trait mapping, candidate gene discovery, and marker development has become more precise and reliable [29]. This article reports refinement of the existing low-density and the first simple sequence repeats (SSR) based genetic map by deploying 58K SNPs “Axiom_*Arachis*” array and identification of QTLs associated with drought tolerance- and ID tolerance-related traits. Further, the candidate genes were identified from QTL regions for drought tolerance and ID tolerance followed by checking their tissue-specific expression.

## 2. Material and Methods

### 2.1. Phenotyping for Drought Tolerance- and Iron Deficiency (ID) Tolerance-Related Traits

One recombinant inbred line (RIL) population derived from the cross TAG 24 × ICGV 86031 was phenotyped for drought tolerance-related traits [16,18,21,22] earlier at ICRISAT-Patancheru (India) (PT) during the post-rainy season in 2004 and 2005. The traits phenotyped included transpiration, transpiration efficiency, specific leaf area (SLA), leaf area (LA), canopy conductance (ISC), delta biomass, shoot biomass, dry weight (DW) increase, total dry matter (TDM), and soil plant analytical development (SPAD) chlorophyll meter reading (SCMR) [30]. Subsequently, the population was phenotyped for transpiration, transpiration efficiency, haulm weight, pod weight, and shoot dry weight under well-watered and water-stressed regimes during the post-rainy season (2007–2008) at ICRISAT-Patancheru (India) (PT) [21]. During 2014 and 2015, the population was phenotyped at ICRISAT-Patancheru (India) (PT) for days to 50% flowering, pod weight, 100 seed weight, haulm weight, and days to maturity. This population was also evaluated at Sadore (SD) (Niger) during 2009 and 2010, and at Bambey (BM) (Senegal) during 2009 for number of primary branches, plant height, SPAD chlorophyll meter reading (SCMR), pod weight, haulm weight, harvest index, shelling percentage (SP), 100 kernels weight (100 KW), and percentage of sound mature kernels (SMK %) under well-watered (WW) and water stress (WS) conditions [22]. Finally, the population was also evaluated for ID tolerance-related traits, namely, visual chlorosis rating (VCR) and stability of soil plant analytical development (SPAD) chlorophyll meter reading (SCMR) at Vijayapur (VJ), UAS-Dharwad (India), during the rainy season of 2013 and 2014 [24,31]. The detailed information on multi-season and multi-location phenotyping data on the above-mentioned traits is summarized in Table 1.

### 2.2. DNA Extraction and Genotyping with 58K SNPs “Axiom_Arachis” Array

Leaf samples were collected for DNA isolation from 25–30-day-old plants from the entire RIL population (309 lines) along with two parental genotypes. The DNA was isolated using a Nucleospin Plant II kit (Macherey-Nagel, Düren, Germany; https://guest.link/UM6). For each sample, ≈100 mg tender leaf tissue was homogenized using 500 μL lysis buffer. A total of 10 μL of RNase solution was mixed in to remove RNA impurities. The mixture was incubated for 1h at 65 °C in a water bath and centrifuged for 20 min at 6000 rpm (revolutions per minute) to collect the clear supernatant. The supernatant was mixed with 450 μL binding buffer. This lysate was filtered through a Nucleospin plant MN column and the column was centrifuged for 1 min at 6000 rpm and discarded the flow-through. Further, the pellet was washed with 400 μL buffer PW1 and centrifuged for 1 min at 6000 rpm. Again, 700 μL buffer PW2 was added to the column and centrifuged for 1 min at 6000 rpm and discarded the flow-through. Finally, 50 μL warm elution buffer (65 °C) was added onto the membrane filter of column and incubated for 5 min at 65 °C and centrifuged for 1 min at 6000 rpm to elute the DNA. The DNA was quantified using a Nanodrop 8000 Spectrophotometer (Thermo Fisher Scientific Inc., Waltham, MA, USA) and the DNA quality was checked on 0.8% agarose gel. Affymetrix “Axiom_*Arachis*” array platform was used to genotype the RIL population with the 58K SNPs, as explained in Pandey et al. [25].

### 2.3. SNP Allele Calling, Filtering, and Quality Control

We used “Best Practice” workflow to perform quality control (QC) analysis of samples to select only those that passed the QC test for further analysis. Best practice workflow checks quality control of samples and plates. The only genotypes passing the thresholds in the Best Practice workflow were processed for performing all the downstream analysis. The “Sample QC” workflow was then used for the set of genotypes passing the QC from Best Practice workflow. The “Genotyping” workflow was used to perform genotyping on the imported CEL files (file format generated by Axiom array), regardless of the sample QC matrix. Summary of the data was produced by using “Summary Only” workflow. The genotypic data based on 58,233 SNPs on 309 RILs were extracted from Axiom Analysis Suite. The SNPs were renamed as chromosome name followed by physical position such as A06_105402882. The chi-square (χ^2^) values calculated for each SNP and SSR marker were used to determine the goodness of fit to the expected 1:1 segregation ratio; highly distorted markers were filtered out for maintaining the quality of genetic map.

### 2.4. Construction of Genetic Map

JoinMap version 4 [32] was used for construction of genetic map. The grouping and ordering of markers were carried out using a regression mapping algorithm. Kosambi’s mapping function was used for converting the recombination frequency into map distance in centiMorgan (cM). The markers were ordered in 20 linkage groups by applying the LOD score (logarithm of the odds) with LOD threshold ranging from 3 to 10 with minimum recombination frequency threshold (∂) of 50%. MapChart software was used to visualize the final charts with locus and position [33].

### 2.5. Identification of Major Main-Effect and Epistatic QTLs

Phenotyping data for drought tolerance-related traits under well-watered and water-stressed regimes recorded at ICRISAT-Patancheru, India (post-rainy season 2004, 2005, 2008, 2014, and 2015); Sadore, Niger (2009 and 2010); and Bambey, Senegal (2009) and the phenotyping data for ID tolerance-related traits generated at Vijayapur, UAS-Dharwad, India (rainy season of 2013 and 2014), were used together with a refined genetic map and genotypic data for QTL analysis. Inclusive composite interval mapping additive (ICIM-ADD) method was used to identify major-effect QTLs and inclusive composite interval mapping epistatic (ICIM-EPI) method was used for identification of epistatic QTLs. Both ICIM-ADD and ICIM-EPI methods were implemented in ICIM software v4.1.0.0. [34]. In ICIM, the *p*-values for entering variables (PIN) and removing variables (POUT) were set at 0.001 and 0.002, respectively, and the scanning step was 1.0 cM. LOD threshold value of 3.0 was used to declare the presence of a QTL. QTLs with >10% phenotypic variance explained (PVE) were considered as major QTLs; the rest were considered as minor QTLs. The name of QTL starts with a lower case “q” followed by abbreviated capital letters to designate the respective trait and chromosome number. If more than one QTL for the same trait was identified, we added another number on the basis of the relative position of QTLs on the chromosome. For example, the *qHW-A01.1* means first QTL identified for haulm weight on chromosome A01.

## 3. Results

### 3.1. Refined Genetic Map on RIL Population TAG 24 × ICGV 86031

Genotypic data generated on RIL population using SNP array with 58,233 SNPs along with earlier mapped 191 SSR loci were subjected to filtering by removing markers with monomorphism and >80% missing data. After very rigorous filtration, 1320 polymorphic SNPs were selected out of the total genotyping data generated for 58,233 SNP loci. Further, 115 SNP and 14 SSR markers showing segregation distortion on the basis of chi-square (χ^2^) test were removed from further analysis. After the stringent filtration, a set of 1205 SNPs and 177 SSRs were used for construction of improved genetic map. As a result, the new genetic map was developed with 1205 loci (1028 SNPs and 177 SSRs) mapped on 20 linkage groups with a total map length of 2598.3 cM. A total of 499 SNP loci and 149 SSRs were mapped onto A-subgenome spanning 1443.4 cM, whereas 529 SNP and 28 SSR loci were mapped onto B-subgenome covering 1154.9 cM map length. Very few loci (20 SNPs) were mapped on B10 with a map length of 57.9 cM and average map distance of 2.9 cM. Maximum number of loci (105) including 86 SNPs and 19 SSRs were mapped on chromosome A06, having a map length of 171.2 cM with average map distance of 1.63 cM. The highest average map distance of 1.5 cM per locus was observed on A09, while A10 had the lowest average map distance with average marker spacing of 6.2 cM (Table 2). Recombination breakpoints across the population for 1205 marker loci are presented in Figure 1a. The map charts showing the positions of marker loci on 20 chromosomes are presented in Figure 1b.

The recombination frequencies between the loci of homologous chromosomes were similar because of genome similarity between both subgenomes (Figure 2a). The genetic map showed collinearity with reference genomes of progenitor species [35], namely, *Arachis duranensis* and *Arachis ipaensis* according to Spearman’s rank correlation. The order of SNP loci between genetic map (cM) and physical map (Mb) was the same as that accessed and visualized using the circos plot (Figure 2b).

### 3.2. Genome-Wide Main-Effect QTLs for Drought Tolerance- and Iron Deficiency (ID) Tolerance-Related Traits

The genome-wide discovery of main-effect QTLs using phenotypic data together with genetic map information and genotyping data identified a total of 133 main-effect QTLs, including 129 QTLs for drought tolerance- and four QTLs for ID tolerance-related traits (Table 3 and Appendix A). Of these 133 QTLs, 19 were major main-effect QTLs (16 QTLs for drought tolerance and 3 QTLs for ID tolerance) detected under water-stressed (WS) and well-watered (WW) conditions (Table 4, Figure 3a) with >10% PVE (Table 4 and Appendix A).

#### 3.2.1. Main-Effect QTLs for Drought Tolerance-Related Traits

Of the 129 main-effect QTLs for drought tolerance-related traits, 6 QTLs were for canopy conductance under WW, 11 QTLs (10 WW and 1 WS) for SCMR, 23 QTLs (16 WW and 7 WS) for leaf area, 13 QTLs (3 WW and 10 WS) for transpiration rate, 13 QTLs (11 WW and 2 WS) for transpiration efficiency, 3 QTLs (WW) for days to 50% flowering, 6 QTLs (WW) for total dry matter, 22 QTLs (16 WW and 6 WS) for haulm weight, 1 QTL (WW) for harvest index, 7 QTLs (6 WW and 1 WS) for seed weight, 2 QTLs (WW) each for number of branches and shelling %, 9 QTLs (4 WW and 5 WS) for pod weight, and 11 QTLs (9 WW and 2 WS) for shoot dry weight (Table 3 and Appendix A).

For drought tolerance-related traits, a total of 16 QTLs with major phenotypic effect were identified for 9 traits (Table 4, Figure 3a). Four QTLs (*qHW-A01.1*, *qHW-A05.4, qHW-B01.4,* and *qHW-B09.1*) associated with haulm weight were detected on chromosomes A01, A05, B01, and B09, respectively, with 10.8% to 20.1% PVE. The favorable alleles for QTLs *qHW-A01.1* and *qHW-A05.4* were contributed by parent ICGV 86031, while favorable alleles for QTLs *qHW-B01.4* and q*HW-B09.1* were contributed by parent TAG 24. Moreover, the QTL *qHW-A05.4* identified on chromosome A05 (16.3% PVE with LOD 11.6), q*HW-B01.4* on chromosome B01 (10.8% PVE with LOD 7.0), q*HW-B09.1* on chromosome B09 (12% PVE with LOD 4.4), and q*HW-A01.1* on chromosome A01 (PVE 20.1% with LOD 13.7) were found promising for further investigation.

Three QTLs were detected for pod weight (*qPW-A03.1*, *qPW-A02.1*, and *qPW-A03.2*) under WS and WW regimes with PVE ranging from 10–14.2% with positive contribution by alleles from TAG 24. A genomic region (A03_101625507-A03_25161497) was identified on chromosome A03 harboring two consistent QTLs for pod weight (*qPW-A03.1* and *qPW-A03.2*) and one QTL for seed weight (*qSW-A03.1*). The QTLs for pod weight (*qPW-A03.1* and *qPW-A03.2*) were detected in both (WW and WS) conditions, which indicated that this QTL region was stable and not essentially affected by water regimes. Moreover, one more QTL (*qPW-A02.1*) was detected on chromosome A02 under WS condition, explaining 10.1% PVE with LOD 3.7. Two major main-effect QTLs (*qDW-A05.3* and *qDW-A05.2*) were identified for dry weight increase and total dry matter on chromosome A05, respectively, with contribution of favorable alleles from parent ICGV 86031. These QTLs explained 10% and 13.9% PVE, with 3.4 and 4.9 LOD, respectively.

Two QTLs (q*SLA-A03.4* and q*SLA-A04.5*) were identified for leaf area on chromosomes A03 (12% PVE and 8.7 LOD) and A04 (16.2% PVE and 10.8 LOD), and the favorable alleles were attributed by the parents TAG 24 (*qSLA-A03.4*) and ICGV 86031 (*qSLA-A04.5*), respectively. One QTL (*qISC-A04.1*) was identified for canopy conductance with 17.2% PVE (LOD of 13.5), and the same region was identified for specific leaf area (*qSLA-A04.5*), explaining 16.2% PVE (10.8 LOD) wherein the favorable allele was contributed by TAG 24. One QTL (*qTR-A09.1*) was identified for transpiration rate on chromosome A09 with 17.3% PVE (LOD 9.7) with favorable allele contributed by TAG 24 (*qTR-A09.1*). Single QTL (q*NB-A07.1*) for number of branches was identified on chromosome A07 explaining 23.3% PVE (LOD 15.5), wherein the favorable allele was contributed by ICGV 86031. One QTL (*qSCMRd-A04.4*) for SCMR under drought stress was identified on chromosome A04, explaining 10.8% PVE (LOD 5.3) with favorable allele contributed by TAG 24. (Table 4, Figure 3a).

A common genomic region for pod weight under WW as well as WS (*qPW-A03.1, qPW-A03.2*) and for seed weight under WW (*qSW-A03.1*) condition was identified on chromosome A03 with 10.0–15.0% PVE. However, higher PVE was recorded under WW condition for both pod and seed weight as compared to WS condition. A common genomic region of main-effect QTLs on chromosome A03 was identified for haulm weight under WW and WS regimes (*qHW-A03.1, qHW-A03.2*), and transpiration efficiency under WW regime (*qTE-A03.1, qTE-A03.2*). A consistent genomic region on chromosome A05 was identified for initial dry weight in 2005 (*qDW-A05.1*), biomass final in 2004 (*qHW-A05.1*), and shoot biomass in 2004 at Patancheru location with a cumulative PVE of 22.1%. A QTL for number of branches in 2009 at Bambey, Senegal (*qNB-A07.1*), was also identified in 2008 at the Patancheru, location in India for transpiration rate observations under WW regimes i.e., TE1, TE2, TE3, and TE4 (*qTE-A07.1*, *qTE-A07.1*, *qTE-A07.1,* and *qTE-A07.4*, respectively). However, higher LOD and PVE were recorded for number of branches at 15.5 and 23.3%, respectively. Small effect QTLs were identified for five traits phenotyped at Patancheru, India, such as total dry matter in 2005 (*qDW-B01.1*), dry weight increase in 2005 (*qDW-B01.2*), pod weight under WW and WS in 2008 (*qPW-B01.1*, *qPW-B01.2*), seed weight under WW and WS (*qSW-B01.1*, *qSW-B01.2*), and transpiration rate in 2005 (*qTR-B01.1*). A QTL region was detected for biomass delta (*qHW-B09.1*) and final biomass (*qHW-B09.2*) in 2004 at Patancheru, India, with PVE 12.0 and 9.8%, respectively.

#### 3.2.2. QTLs for ID Tolerance-Related Traits

Three major main-effect QTLs were identified for ID tolerance- related traits that included SCMR (2 QTLs) and VCR (1 QTL) (Table 4, Figure 3a) in addition to a main-effect QTL, *qSCMR-B07.1,* on B07 with 3.2 LOD and 4.4% PVE (Appendix A). The QTLs for SCMR (*qSCMRID-B03.1* and *qSCMRID-B03.2*) were identified at 60 days after sowing (DAS) and 90 DAS, respectively, for ID tolerance. These QTLs (*qSCMRID-B03.1* and *qSCMRID-B03.2*) identified on chromosome B03 shared the same marker interval B03_13454528 -B03_10796590 at 62 cM for SCMR at 60 DAS and 90 DAS. QTL *qSCMRID-B03.1* explained 22.4% PVE with 4.8 LOD, whereas QTL *qSCMR-B03.2* explained 11% PVE with 4.8 LOD. A single QTL (*qVCR-B03.1*) was identified for VCR on chromosome B03 with 33.9% PVE and 4.4 LOD. A common genomic region on chromosome B03 with higher PVE for SCMR (22.4%) and VCR (33.9%) for ID tolerance (Table 4, Figure 3a) holds promise for further investigation.

### 3.3. Epistatic Interactions (E-QTLs) for Drought Tolerance- and ID Tolerance-Related Traits

A total of 608 E-QTLs including 551 E-QTLs for drought tolerance- and 57 E-QTLs for ID tolerance- related traits were identified with PVE% ranging from 10.0 to 70.9% (Table 3 and Appendix A).

#### 3.3.1. E-QTLs for Drought Tolerance-Related Traits

In total, 551 E-QTLs were identified for 12 drought tolerance-related traits. The highest number of E-QTLs (213) (210 WW and 3 WS) were detected for shoot dry weight (12.1–46.3% PVE and 5.0–20.7 LOD) followed by 166 E-QTLs (158 WW and 8 WS) for haulm weight (16.4–65.1% PVE and 5.0–19.8 LOD) (Table 3 and Appendix A). A total of 71 E-QTLs (1 WW and 69 WS) were detected for shelling percentage (26.9–53.2% PVE and 5.0–15 LOD), 50 E-QTLs under WW for leaf number (18.7–34.9% PVE and 5.1–20.1 LOD), 21 E-QTLs under WS for total dry matter (23.9–66.2% PVE and 5.01–11.1 LOD), 5 E-QTLs under WW for water use efficiency (20.6–32.5% PVE and 5.2–6.1 LOD), 5 E-QTLs (1 WW and 4 WS) for pod weight (10–33.8% PVE and 5.1–5.3 LOD), 3 E-QTLs under WW for transpiration rate (25.8–33.4% PVE and 5.1–5.9 LOD), 1 E-QTL under WW for transpiration efficiency (11.5% PVE and 5.02 LOD), 1 E-QTL under WW days to 50% flowering (11.9% PVE and 5.0 LOD), and 1 E-QTL under WW for SCMR-drought (15.8% PVE and 5.1 LOD).

Four epistatic interactions (E-QTLs) were identified as consistent/common under WW and WS regimes for drought tolerance component traits. For instance, a consistent E-QTL was identified for haulm weight in 2008 under WW (*qqHAULM.11*) and WS (*qqHAULM.3*), and also in 2014 under WS (*qqHAULM.158*) with 19.3–40.0% PVE. One more QTL was identified for haulm weight in 2008 under WW (*qqHAULM.1*) and WS (*qqHAULM.2*) with 32.8% and 34.8% PVE, respectively. Similarly, 1 E-QTL on chromosome A04 was identified for shoot dry weight under WS condition at two different stages (*qqSDW.212*, *qqSDW.213*) with 12.0% PVE. An epistatic QTL for shoot dry weight was detected under WS (*qqSDW.1*) and WW (*qqSDW.94*) with 42.9 and 25.8% PVE, respectively. E-QTLs for most of the component traits of drought were also identified in this study. For instance, an interaction between A06 and B06 was identified for total dry matter, haulm yield, leaf number, and shoot dry weight under WW at various locations including Bambey, Sadore, and Patancheru (Table 5 and Appendix A).

#### 3.3.2. E-QTLs for ID Tolerance-Related Traits

A total of 57 E-QTLs were identified for ID tolerance-related traits including 47 E-QTLs for SCMR with 5–16.6 LOD and 15.8–57.7% PVE, and 10 E-QTLs for VCR with 5–6.8 LOD and 14.6–70.9% PVE (Table 3 and Appendix A).

Altogether, 18 major epistatic QTLs (LOD > 5.0) were identified for component traits VCR (4 QTLs), SCMR (8 QTLs), and haulm weight (6 QTLs). A major E-QTL for VCR (*qqVCR.3*) with the highest PVE of 62.9% and 5.8 LOD showed an interaction between genomic regions of chromosomes B05 and B07. A major E-QTL for SCMR (*qqSCMR.30*) with 49.8% PVE and 6.2 LOD showed an interaction between the genomic regions of chromosomes A03 and B03. A major E-QTL identified for haulm weight (*qqHAULM.5*) on chromosomes B07 and B09 explained 36.7% PVE with 5.5 LOD (Table 5, Figure 3b).

Several E-QTLs were identified as common for ID component traits SCMR and VCR. For instance, a common E-QTL was identified on chromosome A01 for SCMR (*qqSCMR.37*) at 90 DAS and VCR (*qqVCR.1*) at 30 DAS in 2013 with 47.7% and 42.7% PVE, respectively. Moreover, a common E-QTL was detected on chromosome A06 for SCMR (*qqSCMR.6*) and VCR (*qqVCR.9*) at 90 DAS in 2013 with 57.7 and 42.5% PVE, respectively. Similarly, one common E-QTL was identified on chromosome A06 for SCMR at 60 DAS in 2013 (*qqSCMR.6*) and VCR at 30 DAS in 2014 (*qqVCR.9*) with 20.0 and 14.6% PVE, respectively. Most of the common E-QTLs were identified for SCMR at 60 DAS and 90 DAS in 2013 (Appendix A).

### 3.4. Candidate Genes Underlying Major QTL Regions

In this study, 16 QTLs for 9 drought tolerance-related traits, namely, dry weight increase and dry matter production (2 QTLs each), haulm weight (4 QTLs), number of branches (1 QTL), pod weight (3 QTLs), seed weight (1 QTL), SCMR-drought (1 QTL), leaf area (2 QTLs), canopy conductance (1 QTL), transpiration rate (1 QTL), and 3 QTLs for two ID tolerance-related traits—SCMR-ID (2 QTLs) and visual chlorosis rating (1 QTL). The QTLs for haulm weight, SCMR, VCR, seed weight, and pod weight were targeted for identification of candidate genes due to their higher LOD and PVE% (Table 6). The candidate genes and transcription factors (TFs) identified in the major QTL genomic region were involved in various signaling events, acting as key transcriptional activators and saviors of plants during biotic and abiotic stress. The tissue-specific expression of identified candidate genes was studied using gene expression atlas (AhGEA) of *fastigiata* subspecies [36]. Of the 18 candidate genes identified in this study, 11 candidate genes were detected in genes expression atlas across different tissues (Figure 3c). Eight of these candidate genes were identified in a QTL region detected for haulm weight (*qHW-B09.1*) on chromosome B09, including expression of six genes in different tissues during plant growth stages. *MADS-box* gene (*Araip.1IW39*), which encodes transcription factors (TF) with key roles in plant growth and development, was found highly expressed in vegetative and senescence leaves. *Lob* (*lateral organ boundaries*) *domain (Araip.90KYQ)* gene encodes TF, which controls the formation of adventitious crown-root shown high expression in nodules as well as developing pods. *Trehalose phosphate synthase (Araip.UK5PE)* gene that acts as an important modulator for plant development, inflorescence architecture, and important signaling metabolite, regulating carbon assimilation and sugar status, showed highest expression in seed and pod tissues. *Protein phosphatase 2C (Araip.U4R4L)* gene plays a crucial role in abscisic acid (ABA) signaling, biotic and abiotic stress responses, plant immunity, K^+^ nutrient signaling, and plant development, showing high expression in nodules as well as developing pods. *Mate efflux family (Araip.KY5AZ)* genes, which showed high expression in nodules, pods, and flowers, were involved directly or indirectly in mechanisms of disease resistance; nutrient homeostasis (such as Fe^3+^ uptake); and the transport of diverse types of secondary metabolites such as alkaloids, flavonoids, and anthocyanidins, as well as hormones such as ABA, salicylic acid, and auxin. *Glycosyl hydrolase* (GH) *(Araip.4E9NM)* gene involved in cellulose biosynthesis and lignocellulose modification was found in the QTL region on chromosome B09. *Ethylene-responsive transcription factor (Araip.AXK3N)* gene involved in response to various stresses in plants was also detected near the QTL region (*qHW-B09.1*) on chromosome B09.

The *Microtubule-associated protein (Aradu.XX57T)* gene involved in plant morphogenesis, plant hormone signaling, stress, and pathogen response and development of specific morphological structures was found linked to *qPW-A03.1* and *qPW-A03.2* QTLs on chromosome A03. The interaction of *glutathione S-transferase (Aradu.444VJ)* gene and *late embryogenesis abundant (LEA) protein (Aradu.TW8M6)* gene was detected in the QTL mapped on chromosome A03. Similarly, the *bHLH* (*basic helix-loop-helix*) *transcription factor (Aradu.YPV42)* gene involved in controlling grain length and weight was found linked with the QTL region identified on chromosome A03 for pod weight and seed weight.

The *LRR* (*Leucine-rich repeat*)*receptor (Araip.8EV61)* genes constitute the largest group among the receptor-like kinases, and their expression induced by ABA, dehydration, high salt, and low temperature were found near to the peak on chromosome B03 (*qSCMRID-B03.1, qSCMRID-B03.2)*. Seven genes were detected in a QTL region on chromosome B03 (*qVCR-B03.1, qSCMRID-B03.1,* and *qSCMRID-B03.2*) for ID tolerance. Of these seven genes, *dnaJ-related chaperone protein (Araip.5YM5M)* gene, which is an essential molecular chaperone in protein homeostasis and protein complex stabilization under stress conditions, chloroplast development, phototropin-mediated chloroplast movement, and protein import and translocation, showed high expression in leaves during senescence. The *myeloblastosis (MYB) transcription factor (Araip.E5CWX)* gene, which regulates phenylpropanoid pathway genes in response to Fe deficiency, was highly expressed in developing seed. In plants, the *NAC* (*NAM, ATAF,* and *CUC*) *(Araip.PW8UQ)* gene family is one of the largest plant-specific transcription factor families and is involved in various plant processes, including plant development, cell division, functional role in leaf senescence, and stress responses; it was found to be linked to *qSCMRID-B03.1* and *qSCMR-B03.2* QTLs on chromosome B03. High expression of *NAC* TF genes was recorded in senescence leaves, nodules, and developing seed in gene expression atlas. The *2-oxoglutarate/Fe (II)-dependent dioxygenases (Araip.IJN8L)* gene was detected in the QTL region on chromosome B03 for ID tolerance and this gene encodes a family of enzymes involved in various oxygenation/hydroxylation reactions, including the biosynthesis of Fe^3+^-chelating coumarin esculetin Fe uptake in the rhizosphere to maintain Fe homeostasis in plants. *PIP2-5 aquaporin (Araip.K65JZ)* and *ROP (Rho of plants*) *guanine nucleotide exchange factor (Araip.VH7YZ)* genes that relate to various forms of abiotic stress showed high expression in leaves, seeds, and pods tissues (Table 6 and Appendix A, Figure 3c).

## 4. Discussion

Constantly changing climatic conditions around the world demand continuous efforts to understand and adapt to environmental challenges for sustainable crop production. It is therefore essential to develop climate-smart crops in order to assure global food security. Significant progress has been made in the last decade in plant breeding by transferring and pyramiding QTLs controlling the yield attributing traits under drought using marker-assisted selection (MAS) in other crops [37]. In the case of groundnut, although the molecular breeding approach has been used to develop superior lines for oleic acid [38,39] and foliar disease resistance [40], such efforts have not been undertaken for drought tolerance or iron deficiency tolerance due to unavailability of the diagnostic markers and candidate genes. Identification of genomic regions and candidate genes controlling drought tolerance is essential to develop linked markers for use in crop breeding. In groundnut, the genetic basis underlying drought-related traits remains unclear from the time of its domestication. Iron-deficient soil is common worldwide among crops grown in calcareous and alkaline soils due to lower levels of available Fe (Fe^2+^) for uptake. Iron deficiency is a major production constraint in the groundnut-growing areas of several states in India [8], northern China [41], and Pakistan [42], causing a significant reduction in yield. Further, acute iron deficiency results in crop failure and plant death, as iron is a co-factor for thousands of enzymes in plants. Identification and development of iron uptake-efficient genotypes are difficult because of spatial-temporal variability of iron content in the fields [43]. Several factors such as variation in soil type, soil moisture content, soil temperature, and bicarbonate ion concentration in soil are some of the reasons for inconsistent iron deficiency symptoms. For the development of iron uptake-efficient genotypes, it is important to understand the genetic basis of iron deficiency and its component traits under multiple environment trials and soil types. Hence, an RIL population (TAG 24 × ICGV 86031) was used in this study to explore stable QTLs governing drought tolerance and iron deficiency tolerance in groundnut. Genomics-assisted breeding can accelerate the process of developing improved varieties through facilitating marker-based selection of QTLs for target traits [15,44]. Due to large genome size and low genetic diversity in the cultivated gene pool in groundnut, dense genetic maps are difficult to develop for performing high resolution trait mapping. To overcome the problems mentioned above to some extent, high-density SNP array with uniform genome coverage provides opportunity for performing high-resolution trait mapping. The deployment of the “Axiom_*Arachis*” array with 58,233 SNPs [25] in groundnut facilitated generation of high-throughput genotyping data for high-resolution trait mapping [45]. The studies also showed that SLAF-seq can help developing denser genetic map than GBS or SNP array [46,47,48].

This study improved the map density of genetic map for RIL population (TAG 24 × ICGV 86031) to 1205 mapped loci as compared to the first (135 SSR loci, [16]) and second (191 loci, [21]) version of genetic maps. The 1205 mapped loci (1028 SNPs and 177 SSRs) were mapped on 20 linkage groups with a total length of 2598.3 cM and average 2.2 cM distance between loci as compared to much lower average distance between loci (9.34 cM) achieved previously with SSR markers [21]. Therefore, the improved genetic map could discover fine QTL regions for drought tolerance- and iron deficiency tolerance-related traits. In addition to this, the SNPs on 58K SNP array were extracted using well-annotated diploid genomes [35], which makes easy to dive into the genome for functional characterization of identified SNPs and genes. Therefore, we could discover important candidate genes for haulm weight, pod weight, and iron deficiency-related traits. SNP arrays can produce on an average of 96% and maximum of 99% call rates, whereas GBS or ddRAD-seq produces a maximum of 37% call rates. Hence, GBS or ddRAD-seq are less efficient as compared to SNP arrays due to higher missing call rates. In fact, SNP arrays also have some demerits as they do not allow for the discovery of new SNPs; however, they have uniform genome coverage and recombination rates, whereas GBS may have low and non-uniform genome coverage [49]. In addition, the recent study using SNP array also demonstrated its suitability for deployment in genomic selection for improving complex traits in groundnut [12].

This study identified a total of 133 main-effect QTLs (129 QTLs for 14 drought tolerance- and 4 QTLs for 2 ID tolerance-related traits), out of which 19 QTLs (16 QTLs for 9 drought tolerance and 3 QTLs for 2 ID tolerance) were of major effects with >10% PVE. QTLs identified with small effects also contributed to the genetic control of complex polygenic traits. Small-effect QTLs for complex traits are highly polygenic in nature and have been reported in many crops, including maize [50] and rice [51]. Indeed, the contribution of each locus may be negligible, but the total contribution is usually significant for such complex traits. A total of 10 QTLs were identified on chromosomes A02, A03, A04, A05, A09, and B09 under WS conditions. However, under WW regime, only six QTLs were identified on chromosomes A01, A03, A05, and B01. Four QTLs were found on chromosome A03 under both the hydrological conditions. Although many QTLs for yield-related traits under WS were also reported using the same mapping population upon phenotyping in Niger and Senegal [22], this study provided precise position on the genome and facilitated candidate gene discovery.

In a previous study, the QTLs for ID tolerance-related traits were identified on linkage group (LG) AhXIII (B03) with 31% PVE for VCR; however, the QTL for interrelated trait, SCMR, could not be identified in the same genomic region and with similar PVE [24]. On the other hand, in the present study, we successfully detected three major QTLs (*qVCR-B03.1, qSCMRID-B03.1,* and *qSCMRID-B03.2*) sharing the common marker interval on chromosome B03 (B03_13454528 -B03_10796590) for SCMR and VCR with 22.4% and 33.9%PVE, respectively. This suggested the common region (B03:13454528:10796590) as a major genomic region governing ID tolerance.

The QTL for pod weight (*qPW-A03.1*) under WS was identified on chromosome A03 flanked by markers A03_101625507 and A03_25161497 with favorable allele contribution from parent TAG 24. At the same time, under WW conditions, another QTL for pod weight (*qPW-A03.2*) was identified on the same chromosome A03 flanked by the same marker loci with favorable allele contribution from parent TAG 24. The QTLs under two water regimes remained consistent and flanked by the same markers on the same chromosome, thereby showing the stability in its expression and not essentially affected by environmental factors as reported in rice [52,53]. QTL for dry weight increase, dry matter, leaf area, and canopy conductance coincided with the QTLs detected earlier using the same mapping population [16,18,21]. No consistency for QTLs across years was observed for these traits, as has also been seen in other crops [50,54]. Drought tolerance is a complex trait that is a sum total of several component traits, each with small effect QTLs and large epistatic interactions (E-QTLs) [21]. For both qualitative and quantitative traits, epistatic QTLs are considered to be essential factors that result from gene interaction among the genomic regions [55]. Intriguingly, in the present study, we identified a total of 551 E-QTLs for 12 drought tolerance-related traits and 57 E-QTLs for 2 ID tolerance-related traits. Of these, 18 E-QTLs were identified as major E-QTLs for haulm weight, SCMR, and VCR. Recently, epistatic QTLs of major effect for drought tolerance were discovered and successfully deployed in genomic-assisted breeding for improving drought tolerance in different genetic backgrounds in rice to determine grain yield under drought stress [56,57]. A large number of epistatic interactions with higher phenotypic variance were identified for stem rot resistance in groundnut [58]. In the present study, the majority of E-QTLs showed epistasis between genomic regions of homologous chromosomes of sub-genomes. Major epistatic interactions for SCMR were identified between chromosomes A03/B03, A06/B06, and A07/B07. An E-QTL was identified for both SCMR and VCR interacting between the genomic regions of chromosomes A06 and B06. This indicates that there are some common regulators for SCMR and VCR, the component traits of iron deficiency.

An earlier version of the SSR-based genetic map had larger marker intervals and also sequenced genomes and genome annotations were not available for the groundnut researchers while performing previous studies which hampered in identifying the genomic regions precisely and discovering the genes. The present study made use of available genome sequence and annotations [35] in addition to gene expression atlas [36] for gene discovery in the vicinity of flanking markers of the QTL region. A genomic region of 1.8 Mb housing three genes, namely, *MADS-box*, *glycosyl hydrolase* and *malate dehydrogenase* were found to be linked to haulm weight (*qHW-B09.1*) QTL on chromosome B09. *MADS-box* family genes play a significant role in plant growth and development [59]. One region of 2.3kb (144886665–144889053) on chromosome B09 harbored the *glycosyl hydrolase* gene previously reported for cellulose biosynthesis and lignocellulose modification [60]; high expression levels of GH genes modulate cellulose levels, resulting in high biomass yield [61,62]. The *malate dehydrogenase (Araip.RXS5A)* gene was found located in the ~1.1 Mb region with a flanking SNP (B09_16205676), which is thought to be related to biomass and plant height in maize [63].

The genomic region associated with pod weight and seed weight QTLs located between A03_101625507 and A03_25161497 on chromosome A03 harbored three genes encoding for a *bHLH transcription factor*, *microtubule-associated protein*, and *malate dehydrogenase*. The basic transcription factor, *bHLH* family, is involved in plant growth and developmental processes, as well as in stress responses [64], seed length, and seed weight [65]. The second candidate gene *(Aradu.XX57T)* was found located ~1.7 Mb away, which encodes a *microtubule-associated protein* and reportedly influences seed shape and pod weight [65] by regulating microtubule growth in groundnut. The third encodes for a *malate dehydrogenase* that is a glucose catabolism-related enzyme involved in the deterioration of groundnut seeds during storage. 

Two genes *2-oxoglutarate/Fe(II)-dependent dioxygenases* and *MYB* were identified linked to VCR major QTL. The *2-oxoglutarate/Fe (II)-dependent dioxygenase* family gene is involved in various oxygenation/hydroxylation reactions [66], including the biosynthesis of Fe^3+^-chelating coumarin esculetin, which is released into the rhizosphere for Fe uptake [67] to maintain Fe homeostasis in plants [67]. One region of 1.6kb on chromosome B03 (B03_11843227-B03_11844921) harbors the*MYB transcription factor* gene that regulates phenylpropanoid pathway genes under Fe deficiency or alkaline stress [68,69]. Previous studies reported that MYB 10 and MYB72 are essential TFs for plant growth and development under Fe deficiency conditions [70], which regulates the expression of several genes involved in phenylpropanoid, shikimate, and nicotianamine biosynthesis pathways [71,72]. Under iron deficiency, Palmer et al. [70] reported that MYB10 and MYB72 act as a regulatory cascade to drive the gene expression of nicotianamine synthase genes (*NAS2* and *NAS4*). The 2.6 Mb QTL region on chromosome B03 harboring two main QTLs for SCMR contains the *NAC-domain* transcription factor gene. The *NAC* family gene members are well studied senescence-associated transcription factors [73,74] that perform diverse functions including embryonic, floral, and vegetative development; lateral root formation; as well as tolerance to various biotic and abiotic stressors [75]. In major crops, overexpression or silencing of *NAC* family member genes concedes their roles in yield increase (*OsNAC5*) and drought resistance (*OsSNAC1*) in rice, and grain protein improvement in wheat (*TtNAM-B1*RNAi and *TaNAC-S*) [76,77,78,79].

## 5. Conclusions

A refined genetic map of groundnut was constructed using SSR and SNP markers obtained through 58K SNP “Axiom_*Arachis*” array. This genetic map is composed of 1205 marker loci covering a map distance of 2598.3 cM. On the basis of this genetic map, we identified 19 major QTLs for 11 traits (9 drought tolerance- and 2 ID tolerance-related traits). For iron deficiency tolerance, a 2.6 Mb QTL region on B03 was found to harbor three major-effect QTLs, i.e., one for VCR explaining the highest PVE of 33.9% and two QTLs for SCMR explaining the 11–22.4% PVE. Evaluation of this region with the groundnut genome assembly followed by genome annotations identified important genes and transcription factors such as *NAC domain, 2-oxoglutarate/Fe (II)-dependent dioxygenase*, and *MYB transcription factor.* The major effect QTLs identified for improving drought tolerance-related traits, namely, haulm weight (*qHW-A05.4*), pod weight (*qPW-A03.2*), and 100 seed weight (*qSW-A03.1*), while a major and consistent QTL on B03 for iron deficiency tolerance-related traits, namely, VCR (*qVCR-B03.1)* and SCMR (*qSCMRID-B03.1* and *qSCMRID-B03.2)* hold great promise, after validation, for deployment in groundnut breeding. The genomic regions identified in the present study will be further saturated using fine mapping or sequencing-based trait mapping to discover candidate genes associated with drought and iron deficiency tolerance. The discovered candidate genes will be further deployed for development of diagnostic markers, which can be used to enhance the yield under drought stress and iron deficiency.

## Figures and Tables

**Figure 1 genes-12-00037-f001:**
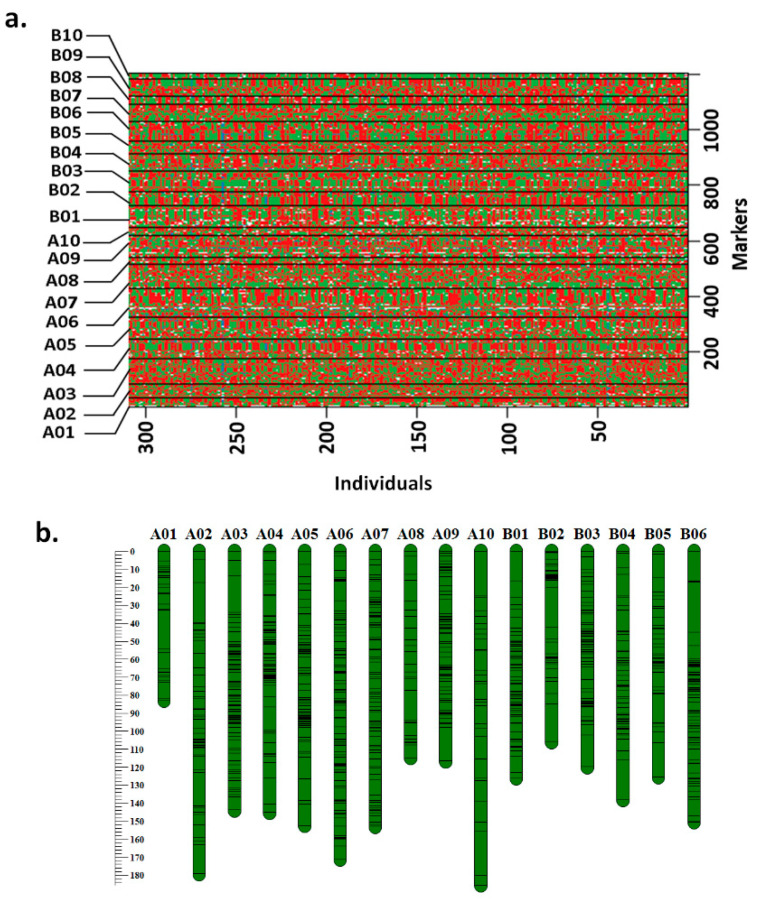
A refined genetic map comprising 1205 loci for TAG 24 × ICGV 86031 RIL population: (**a**) recombination breakpoints between TAG 24 and ICGV 86031; (**b**) map chart of 20 chromosomes.

**Figure 2 genes-12-00037-f002:**
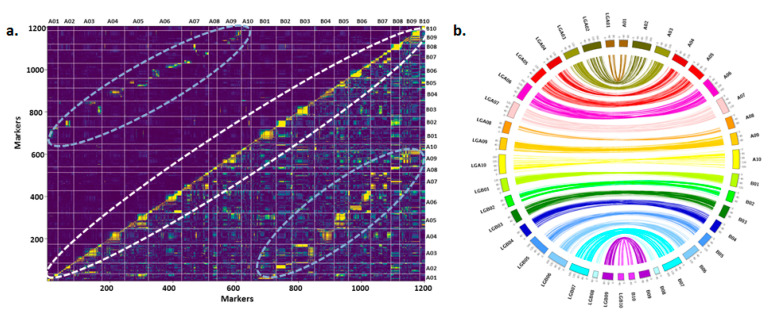
Pair-wise recombination fractions and recombination blocks across 20 chromosomes of *Arachis hypogaea*. (**a**) Recombination frequency plot representing the central unique set of markers segregating on each chromosome. On either side, the markers are showing the similar recombination frequency between homologous chromosomes of ***Arachis***
*ipaensis* and ***Arachis***
*duranensis*. (**b**) Circos plot representing the collinearity of genetic map between reference genomes of both sub-genomes.

**Figure 3 genes-12-00037-f003:**
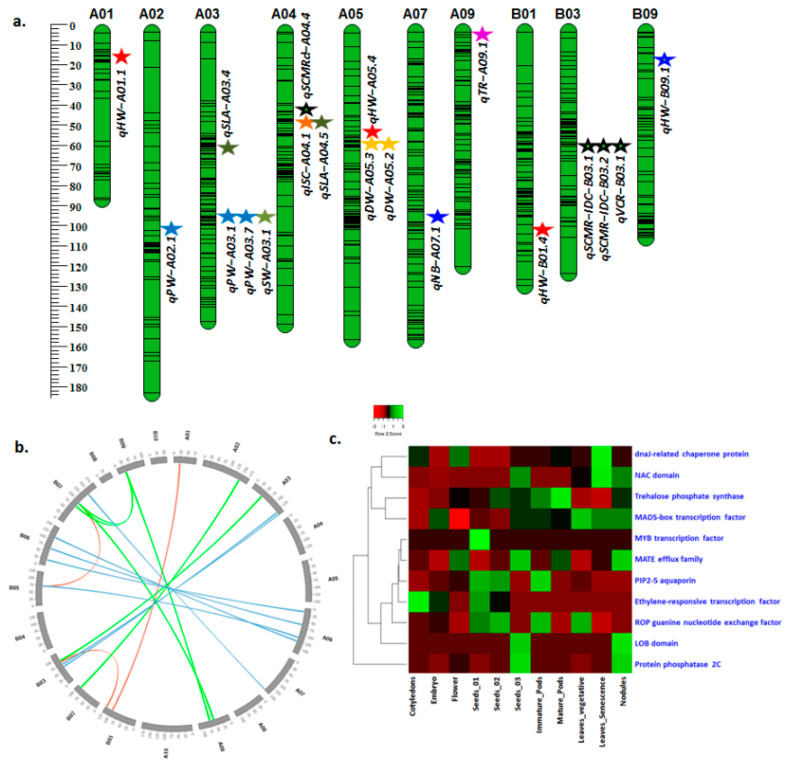
Genomic positions of major main-effect and epistatic-effect QTLs and expression of candidate genes detected in these QTL regions. (**a**) Genomic positions of major effect QTLs for drought tolerance- and iron deficiency (ID) tolerance-related traits on map chart with chromosome number and genetic positions. (**b**) Major epistatic interactions identified for the traits, namely VCR, SCMR, and haulm weight. Circos plot depicts 20 *A. hypogaea* chromosomes and interaction between the genomic regions of these chromosomes. Most of the major epistatic interactions were found in the genomic regions of homologous chromosomes of both sub-genomes *A. ipaensis* and *A. duranensis.* Outer track represents 20 pseudomolecules of groundnut in gray color. Inner lines of three different colors (red—VCR, green—haulm weight, and blue—SCMR) represents the epistatic interactions between genomic regions of 20 pseudomolecules of groundnut. (**c**) Tissue-specific expression of candidate genes identified in QTL regions of SCMR, VCR, pod weight, and haulm weight. Expression of candidate genes was studied using *A. hypogaea* gene expression atlas [36]. The expression of 11 candidate genes in 11 tissues (Cotyledons, Embryo, Flower, Seeds_01, Seeds_02, Seeds_03, Immature_Pods, Mature_Pods, Leaves_vegetative, Leaves_Senescence and, Nodules) is plotted in the heatmap.

**Table 1 genes-12-00037-t001:** Summary of phenotyping data generated for drought tolerance- and iron deficiency tolerance-related traits on TAG 24 × ICGV 86031 recombinant inbred line (RIL) population for 22 traits across four locations and eight years.

Traits	2004	2005	2008	2009	2010	2013	2014	2015
Days to 50% flowering	-	-	-	-	-	-	PT	PT
Dry matter	-	PT (2)	-	-	-	-	-	-
Carbon discrimination ratio (delta13C)	PT	-	-	-	-	-	-	-
Haulm weight	PT (4)	-	PT-WW, PT-WS	SD-WS, SD-WW, BM-WW, BM-WS, PT-WW	SD-WW, SD-WS	PT-WW, PT-WS	PT-WW, PT-WS,	PT-WS, PT-WW
Harvest index	-	-	-	BM-WW, BM-WS, SD-WW, SD-WS,	SD-WW, SD-WS	-	PT-WW, PT-WS	-
100 seed weight	-	-	PT-WW, PT-WS	PT-WW	-	PT-WW, PT-WS	PT-WW, PT-WS	PT-WW, PT-WS
Canopy conductance	PT-WW	PT-WW	-	-	-	-	-	-
Leaf area	PT-WW, PT-WS, PT-H	PT-WW, PT-WS, PT-H	PT-WW, PT-WS	-	SD-WW, SD-WS	-	-	-
Leaf number	-	-	-	-	SD-WW	-	-	-
Days to maturity	-	-	-	BM-WW, BM-WS	-	-	-	-
Number of branches	-	-	-	BM-WW	-	-	-	-
Pod weight	-	-	PT-WW, PT-WS	BM-WW, BM-WS, SD-WW, SD-WS	SD-WW, SD-WS	PT-WW, PT-WS	PT-WW, PT-WS	-
Pod yield	-	-	-	-	-	-	PT-WW	PT-WW
SCMR-drought	-	PT(8)-WW	-	BM-WW, BM-WS, SD-WW, SD-WS	SD-WW, SD-WS	PT-WW, PT-WS	PT-WW, PT-WS	-
Shelling percentage	-	-	-	-	-	PT-WW, PT-WS	PT-WW, PT-WS	PT-WW, PT-WS
Shoot dry weight	-	-	PT(4)-WW, PT(4)-WS	BM-WW, BM-WS	-	-	-	-
Total dry matter	-	PT-WW	-	-	-	-	PT-WS	PT-WS
Transpiration efficiency	PT-WW, PT-WS	PT-WW, PT-WS	PT(4)-WW, PT(4)-WS	-	-	-	-	-
Transpiration rate	PT-WS	PT-WS	PT(4)-WW, PT(4)-WS	-	-	-	-	-
Water use efficiency	PT	-	-	-	-	-	-	-
**Iron Deficiency Tolerance**								
SCMR-ID	-	-	-	-	-	VJ (3)	VJ (3)	-
Visual chlorosis rating (VCR)	-	-	-	-	-	VJ (3)	VJ (3)	-

Patancheru, India: PT; Sadore, Niger: SD; Bambey, Senegal: BM; Vijayapura, India: VJ; well-watered: WW; water-stressed: WS.

**Table 2 genes-12-00037-t002:** Summary of genetic map constructed using single nucleotide polymorphism (SNP) and simple sequence repeat (SSR) markers.

Chr	Total Loci	Total Mapped SNP Loci	SSR Loci	Map Length (cM)	Average Map Distance (cM/ Locus)
A01	33	21	12	82.7	2.5
A02	51	41	10	179.1	3.5
A03	89	73	16	143.7	1.6
A04	73	54	19	145.0	2.0
A05	76	60	16	152.3	2.0
A06	105	86	19	171.2	1.6
A07	86	56	30	152.8	1.8
A08	27	14	13	114.7	4.2
A09	78	68	10	116.4	1.5
A10	30	26	4	185.4	6.2
B01	80	58	22	126.0	1.6
B02	50	50	0	105.8	2.1
B03	73	73	0	119.7	1.6
B04	62	57	5	138.0	2.2
B05	45	45	0	125.4	2.8
B06	71	71	0	150.5	2.1
B07	64	64	0	183.4	2.9
B08	28	28	0	46.1	1.6
B09	64	63	1	102.1	1.6
B10	20	20	0	57.9	2.9
Total	1205	1028	177	2598.3	2.2

**Table 3 genes-12-00037-t003:** Summary of main-effect and epistatic quantitative trait loci (QTLs) for drought tolerance-and iron deficiency tolerance-related traits.

	Main Effect QTLs					Epistatic QTLs (E-QTLs)				
Traits	WW	WS	Total QTLs	LOD Range	PVE Range (%)	WW	WS	Total E-QTL	LOD Range	PVE Range (%)
A. Drought Tolerance-Related Traits										
Days to 50% flowering	3	0	3	3.1–3.9	5.3–6.8	1	0	1	5	11.9
Haulm weight	16	6	22	3.1–13.7	4.3–20.1	158	8	166	5.0–19.8	16.4–65.1
Total dry matter	6	0	6	3.3–5.3	4.8–13.9	0	21	21	5.0–11.1	23.9–66.2
Harvest index	1	0	1	3.5	5.3	-	-	-	-	-
Canopy conductance	6	0	6	3.1–13.5	3.1–17.2	-	-	-	-	-
Leaf number	-	-	-	-	-	50	0	50	5.1–20.1	18.7–34.9
Number of branches	2	0	2	4.8–15.5	6.4–23.3	-	-	-	-	-
Pod weight	9	2	9	3.1–6.5	5.1–14.2	1	4	5	5.1–5.3	10.0–33.8
Transpiration efficiency	11	2	13	3.1–4.8	4.9–8.6	1	0	1	5	11.5
Shelling %	2	0	2	3.1	4.8–5.8	2	69	71	5.0–15.0	26.9–53.2
Shoot dry weight	9	2	11	3.1–6.3	4.2–9.3	210	3	213	5.0–20.7	12.1–46.3
Leaf area	16	7	23	3.1–10.8	5.0–16.2	14	0	14	5.0–8.0	29.0–36.3
Seed weight	6	1	7	3.2–6.4	3.9–15.0	-	-	-	-	-
Transpiration rate	11	2	13	3.2–9.7	4.3–17.3	3	0	3	5.1–5.9	25.8–33.4
SCMR-drought	2	10	11	3.2–5.3	4.8–10.8	1	0	1	5.13	15.8
Water use efficiency			-	-	-	5	0	5	5.2–6.1	20.6–32.5
B. Iron Deficiency Tolerance-Related Traits										
SCMR-ID	2	0	3	3.2–4.8	4.4–22.4	47	0	47	5.0–16.6	15.8–57.7
VCR	2	0	1	4.4	33.9	10	0	10	5.0–6.8	14.6–70.9

WW: well-watered; WS: water-stressed; SCMR: soil plant analytical development (SPAD) chlorophyll meter reading; VCR: visual chlorosis rating; TE: transpiration efficiency; LOD: logarithm of odds; PVE: phenotypic variance explained.

**Table 4 genes-12-00037-t004:** Major main-effect QTLs for drought tolerance- and iron deficiency tolerance-related traits.

Traits	WW/WS	Loc	Year	QTL Name	Chr	Pos (cM)	Left Marker	Right Marker	LOD	PVE (%)	Add
A. Drought Tolerance-Related Traits											
Dry weight increase	WW	PT	2005	*qDW-A05.3*	A05	55	PM375	A05_25039519	3.4	10.0	0.3
Total dry matter	WW	PT	2005	*qDW-A05.2*	A05	55	PM375	A05_25039519	4.9	13.9	0.5
Haulm weight	WW	BM	2009	*qHW-A01.1*	A01	23	Seq13A10	A01_96982501	13.7	20.1	1.5
	WW	BM	2009	*qHW-A05.4*	A05	50	GM2246	A05_25200285	11.6	16.3	1.3
	WW	BM	2009	*qHW-B01.4*	B01	101	B01_134144284	B01_134275884	7.0	10.8	−1.1
Delta biomass	WW	PT	2004	*qHW-B09.1*	B09	22	B09_145215085	B09_16205676	4.4	12.0	−0.2
Canopy conductance	WW	PT	2004	*qISC-A04.1*	A04	49	A04_2540668	Seq19H03	13.5	17.2	−0.1
Number of branches	WW	BM	2009	*qNB-A07.1*	A07	99	IPAHM689	TC1A02	15.5	23.3	0.4
Pod weight	WS	PT	2008	*qPW-A03.1*	A03	92	A03_101625507	A03_25161497	3.7	10.0	−31.1
	WW	SD	2009	*qPW-A02.1*	A02	101	Seq16C06	A02_67418614	3.7	10.1	−6.4
	WW	PT	2008	*qPW-A03.2*	A03	92	A03_101625507	A03_25161497	3.6	14.2	−75.7
SCMR-drought	WW	PT	2004	*qSCMRd-A04.4*	A04	46	GM694	A04_2540668	5.3	10.8	−0.7
Specific leaf area	WW	PT	2004	*qSLA-A03.4*	A03	58	GM660	GM679	8.7	12.0	−1.8
	WW	PT	2004	*qSLA-A04.5*	A04	49	A04_2540668	Seq19H03	10.8	16.2	17.5
100 Seed weight	WW	PT	2008	*qSW-A03.1*	A03	92	A03_101625507	A03_25161497	3.9	15.0	−56.3
Transpiration rate	WS	PT	2008	*qTR-A09.1*	A09	3	A09_117790031	A09_118421381	9.7	17.3	−33.6
B. Iron Deficiency Tolerance-Related Traits											
VCR at 30DAS	WW	VJ	2014	*qVCR-B03.1*	B03	62	B03_13454528	B03_10796590	4.4	33.9	−0.3
SCMR-ID at 60DAS	WW	VJ	2014	*qSCMR-ID-B03.1*	B03	62	B03_13454528	B03_10796590	4.8	22.4	2.9
SCMR-ID at 90DAS	WW	VJ	2014	*qSCMR-ID-B03.2*	B03	62	B03_13454528	B03_10796590	4.8	11.0	2.7

PT: Patancheru, India; SD: Sadore, Niger; BM: Bambey, Senegal; VJ: Vijayapura, India; WW: well-watered; WS: water-stressed; SCMR: SPAD chlorophyll meter reading; VCR: visual chlorosis rating; LOD: logarithm of odds; PVE: phenotypic variance explained; Loc: location; Chr: chromosome; Pos: position; cM: centiMorgan; Add: additive effect; ID: iron deficiency.

**Table 5 genes-12-00037-t005:** Major epistatic QTLs for drought tolerance- and iron deficiency tolerance-related traits.

Trait	WW/WS	Loc	Year	QTL(s)_Name	Chr 1	Pos 1 (cM)	Left Flanking Marker_1	Right Flanking Marker_1	Chr 2	Pos 2 (cM)	Left Flanking Marker_2	Right Flanking Marker_2	LOD	PVE(%)
A. Drought Tolerance-Related Traits														
Haulm weight	WW	PT	2008	*qqHAULM.5*	B07	75	B07_1945835	B07_1907545	B09	25	Seq19B01	B09_16205676	5.49	36.7
	WW	PT	2008	*qqHAULM.6*	B07	90	B07_995898	B07_578122	B09	25	Seq19B01	B09_16205676	5.02	35.2
	WW	PT	2008	*qqHAULM.7*	A09	55	A09_29837655	A09_14222907	B07	90	B07_995898	B07_578122	6.07	34.8
	WW	PT	2008	*qqHAULM.9*	A09	70	A09_113636594	A09_8787029	B09	25	Seq19B01	B09_16205676	5.79	26.2
	WW	PT	2008	*qqHAULM.10*	A02	145	A02_88924681	A02_89185231	B02	90	B02_99720342	B02_101402317	5.11	25.6
	WW	PT	2008	*qqHAULM.11*	A03	45	A03_131260373	IPAHM177	B03	85	B03_132031566	B03_22451302	5.2	21.9
SCMR-drought	WW	PT	2004	*qqSCMR.48*	A06	110	A06_10555841	A06_14851576	B05	70	B05_122054913	B05_148647711	5.13	15.8
B. Iron Deficiency Tolerance-Related Traits														
SCMR-ID at 60DAS	WW	VJ	2013	*qqSCMR.1*	A06	10	A06_105316360	A06_110918357	B06	20	B06_135787232	B06_12479005	6.08	25.7
	WW	VJ	2013	*qqSCMR.2*	A07	150	A07_61889239	A07_68857206	B07	140	B07_1924101	B07_31546304	5.41	23.9
	WW	VJ	2013	*qqSCMR.3*	A03	130	A03_116829847	A03_16185359	B03	60	B03_18642606	B03_13454528	5.97	20.8
	WW	VJ	2013	*qqSCMR.4*	A06	125	A06_92969207	A06_97639398	B06	110	B06_112293470	B06_123620597	7.9	20.0
	WW	VJ	2013	*qqSCMR.5*	A06	60	A06_70533409	A06_80763443	B06	65	B06_87676700	B06_62922699	5.11	19.1
SCMR-ID at 90DAS	WW	VJ	2013	*qqSCMR.30*	A03	140	A03_121816921	A03_7178082	B03	70	B03_135542931	B03_11838056	6.28	49.7
	WW	VJ	2013	*qqSCMR.47*	A03	130	A03_116829847	A03_16185359	B03	60	B03_18642606	B03_13454528	5.09	29.2
VCR at 30DAS	WW	VJ	2014	*qqVCR.3*	B05	70	B05_122054913	B05_148647711	B07	90	B07_995898	B07_578122	5.75	62.9
	WW	VJ	2014	*qqVCR.4*	B01	120	B01_133569092	B01_3669866	B03	80	B03_29375836	B03_5906274	5.3	32.4
	WW	VJ	2014	*qqVCR.5*	A01	20	A01_65424740	A01_105900135	B01	95	IPAHM569	B01_114422004	5.02	15.7
	WW	VJ	2014	*qqVCR.6*	A06	125	A06_92969207	A06_97639398	B06	110	B06_112293470	B06_123620597	5.42	14.6

**Table 6 genes-12-00037-t006:** Candidate genes underlying QTLs for drought tolerance- and ID tolerance-related traits.

Traits	QTL Name	Gene Location	Gene Model	Nearest SNP (bp)	Functional Annotation
A. Drought Tolerance-Related Traits					
Haulm weight	*qHW-B09.1*	Araip.B09	Araip.1IW39	145215085	MADS-box transcription factor
		Araip.B09	Araip.4E9NM	145215085	Glycosyl hydrolase
		Araip.B09	Araip.RXS5A	16205676	Malate dehydrogenase
		Araip.B09	Araip.90KYQ	16205676	LOB domain
		Araip.B09	Araip.UK5PE	145215085	Trehalose phosphate synthase
		Araip.B09	Araip.U4R4L	16205676	Protein phosphatase 2C
		Araip.B09	Araip.KY5AZ	16205676	MATE efflux family
		Araip.B09	Araip.AXK3N	145215085	Ethylene-responsive transcription factor
Pod weight Seed weight	*qPW-A03.1* *qPW-A03.2* *qSW-A03.1*	Aradu.A03	Aradu.YPV42	25161497	bHLH transcription factor
		Aradu.A03	Aradu.TW8M6	101625507	Late embryogenesis abundant (LEA) protein
		Aradu.A03	Aradu.XX57T	101625507	Microtubule-associated protein
B. Iron Deficiency Tolerance-Related Traits					
SCMR VCR	*qSCMR-ID-B03.1* *qSCMR-ID-B03.2*	Araip.B03	Araip.PW8UQ	10796590	NAC domain
		Araip.B03	Araip.8EV61	13454528	LRR receptor
		Araip.B03	Araip.5YM5M	13454528	dnaJ-related chaperone protein
		Araip.B03	Araip.K65JZ	10796590	PIP2-5 aquaporin
		Araip.B03	Araip.VH7YZ	13454528	ROP guanine nucleotide exchange factor
	*qVCR-B03.1*	Araip.B03	Araip.IJN8L	10796590	2-Oxoglutarate/Fe(II)-dependent dioxygenase
		Araip.B03	Araip.E5CWX	13454528	MYB transcription factor

SCMR: SPAD chlorophyll meter reading; VCR: visual chlorosis rating.

## Data Availability

The data and detailed results are provided in Appendix A.

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
