# Peer review of "Improved Genetic Map Identified Major QTLs for Drought Tolerance- and Iron Deficiency Tolerance-Related Traits in Groundnut"

_genes, 2020, doi:10.3390/genes12010037_

Round 1

Reviewer 1 Report

MAJOR SUBSTANTIVE REMARKS

The main content-related remark concerns the terminology and classification of phenotypic features.

Term “iron-deficiency chlorosis tolerance” / “IDC tolerance” seems incorrect. Chlorosis is caused by iron deficiency. Plants can tolerate iron deficiency, not chlorosis. The degree of chlorosis is a symptom and can be a measure of this tolerance. Therefore, I suggest to use term “iron-deficiency tolerance” / “ID tolerance”

Also the separation of “iron-deficiency chlorosis tolerance” features as a separate category seems unjustified. Especially that only three features are included in this category, which in fact are three measurements of the same parameter - chlorophyll content (chlorosis). This is evidenced by the detection of a common QTLs (qVCR-B03.1, qSCMRidc-B03.1 and qSCMRidc-B03.2). In fact, you detected not only “the common marker interval” (Discussion, line 80) but, most likely, also the same gene. Therefore the next statement from the Discussion section (lines 106-107) “This indicates that there are some common regulators for SCMR and VCR the component traits of IDC.” is obvious and indisputable.

Taking into account the above also terms “IDC related SCMR” and “VCR traits” are incorrect.

If you do not select the “iron-deficiency chlorosis tolerance” class, you will not need to include the remaining features in the group “drought tolerance related traits”.

  • Additionally, term “drought tolerance related traits” should be explained. You should state how it is known that analyzed features are drought dependent. Was this already stated in previous studies? If not or if not all features were proved to be dependent on water deficits, maybe you should consider using the phrase "traits potentially related to drought tolerance"
  • Taking into account the above comments - it is not understandable, why the parameter SCMR (SPAD chlorophyll meter reading) is distinguished into two categories (“SCMR-drought” and “SCMR- Iron deficiency chlorosis”). Regardless of the conditions and location of experiment, it relates to the same feature - the relative content of chlorophyll measured by SPAD meter.

REQUIRED / SUGGESTED MINOR CORRECTIONS:

  • I suggest you consider preparing a list of abbreviations
  • The keywords should not contain the words contained in the title. Instead of repeating words, the keywords should contain:
    • the Latin name of the species
    • the usual name of the species, common synonymous to groundnut (peanut)
    • “abiotic stress” instead of “drought tolerance, iron deficiency chlorosis tolerance”
  • Formulation “map density”
    • Abstract; line marked as line 50: parameter “2.2 cM/locus” is not an “average marker density” or “map density”. You should use the term “average distance between loci” (2.2 cM). Density can be referred to as the number of loci per (e.g.) 1 cM or 10 cM.
    • Table 2 “map distance”->”map length”, “map density” -> “average map distance”
    • “However, the majority of these studies failed to provide conclusive results due to a lack of optimum map density ( 56-347 loci).” -> However, the majority of these studies failed to provide conclusive results due to a lack of sufficiently extended genetic map (∼ 56-347 loci)
  • Section1. Phenotyping for drought and iron deficiency chlorosis (IDC) tolerance related traits, lines 9-14: “Later this population was also evaluated at Sadore (SD) (Niger) during 2009 and 2010 […]”.The previous sentence lists the years 2014-2015. 2009-10 is not “later” than 2014-15.
  • Section 2. DNA extraction and genotyping with 58K SNPs ‘Axiom_Arachis’ array: It is enough to mention, that the DNA isolation was carried out in accordance with the manufacturer's instructions, or indicate only modifications (if any). The details in lines 5 – 15 are not needed.
  • Among the references (e.g. in the discussion, next to items 46 and 47) there should be a work cited:

Hu, X.H., Zhang, S.Z., Miao, H.R. et al. High-Density Genetic Map Construction and Identification of QTLs Controlling Oleic and Linoleic Acid in Peanut using SLAF-seq and SSRs. Sci Rep 8, 5479 (2018). https://doi.org/10.1038/s41598-018-23873-7,

which concerns high density map of  the same species, refers to SLAF markers and QTLs of important traits.

  • Discussion, line marked as 24: […] molecular breeding approach has been used to develop superior lines for oleic acid [38, 39] and foliar disease [40], such efforts have not been made in drought or IDC […] -> […] molecular breeding approach has been used to develop superior lines for oleic acid [38, 39] and foliar disease resistance [40], such efforts have not been undertaken for drought resistance or iron deficits resistance […]
  • Discussion, line marked as 33,36: […] iron efficient genotypes […] -> […] iron uptake- efficient genotypes […]
  • Discussion, line marked as 85: Instead of "However [...]" -> "At the same time […]" would be more appropriate to start this sentence.
  • Discussion, line marked as 96: "Surprisingly [...]".This requires an explanation of what the surprise is about. Surprisingly much?
  • Conclusions, line marked as 158: “The major effect QTLs identified for improving drought tolerance related traits namely haulm weight (qHW-A05.1)”. There is no QTL named “qHW-A05.1” in the main manuscript, only in Suplemmentary Table 2. Do you mean “qHW-A05.4”?

MINOR NOTEs of EDITORIAL NATURE  

  • Section “Material and Methods” is numbered as “1” instead “2”
  • Section 2.1. Main-effect QTLs for drought tolerance related traits: paragraph 2, line 2: Table 3 -> Table 4 (incorrect reference)
  • Table 3: all traits have full names, but TE. Expand abbreviation “TE
  • Figure 3: qPW-A03.7 -> qPW-A03.2
  • Discussion, line marked as 128: catabolismrelated -> catabolism related
  • Discussion, line marked as 129: enzymeinvolved -> enzyme involved
  • Discussion, line marked as 140: andnicotian amine -> and nicotinamine
  • Discussion, line marked as 141-142 “Palmer et al. [69] reported under Fe deficiency in regulating the gene expression of nicotianaminesynthase genes” - grammar ?

Author Response

Response to the comments of the Reviewer#1

MAJOR SUBSTANTIVE REMARKS

The main content-related remark concerns the terminology and classification of phenotypic features.

Term “iron-deficiency chlorosis tolerance” / “IDC tolerance” seems incorrect. Chlorosis is caused by iron deficiency. Plants can tolerate iron deficiency, not chlorosis. The degree of chlorosis is a symptom and can be a measure of this tolerance. Therefore, I suggest to use term “iron-deficiency tolerance” / “ID tolerance”

Also the separation of “iron-deficiency chlorosis tolerance” features as a separate category seems unjustified. Especially that only three features are included in this category, which in fact are three measurements of the same parameter - chlorophyll content (chlorosis). This is evidenced by the detection of a common QTLs (qVCR-B03.1, qSCMRidc-B03.1 and qSCMRidc-B03.2). In fact, you detected not only “the common marker interval” (Discussion, line 80) but, most likely, also the same gene. Therefore the next statement from the Discussion section (lines 106-107) “This indicates that there are some common regulators for SCMR and VCR the component traits of IDC.” is obvious and indisputable.

Taking into account the above also terms “IDC related SCMR” and “VCR traits” are incorrect

If you do not select the “iron-deficiency chlorosis tolerance” class, you will not need to include the remaining features in the group “drought tolerance related traits”.

Authors’ Response: We thank the Reviewer for this important suggestion. We have now used suggested term “iron deficiency (ID)” in place of “iron deficiency chlorosis (IDC)” throughout the MS.

SCMR-ID (SPAD Chlorophyll Meter Reading –leaf chlorosis) and VCR (Visual chlorosis rating) are two different scales to measure the leaf chlorosis in iron deficient soils. There is a negative correlation between SCMR and VCR and that is may be the reason behind the common QTL for SCMR and VCR on chromosome B03. However, looking at the additive effects of these QTLs, it is negative for VCR (Add= -0.3) contributed by TAG 24 (Susceptible to iron deficiency) and positive for SCMR at 60 DAS (Add=2.9) and SCMR at 90 DAS (Add=2.7) contributed by iron deficiency tolerant parent ICGV 86031.   

  • Additionally, term “drought tolerance related traits” should be explained. You should state how it is known that analyzed features are drought dependent. Was this already stated in previous studies? If not or if not all features were proved to be dependent on water deficits, maybe you should consider using the phrase "traits potentially related to drought tolerance"

Authors’ Response: Thanks for this important suggestion. Similar traits under the drought tolerance related traits were studied in previous groundnut studies (Varshney et al. 2009 [16], Gautami et al. 2012 [18], Ravi et al. 2011 [21], Faye et al. 2015 [22]) and these references were cited in MS.

  • Taking into account the above comments - it is not understandable, why the parameter SCMR (SPAD chlorophyll meter reading) is distinguished into two categories (“SCMR-drought” and “SCMR- Iron deficiency chlorosis”). Regardless of the conditions and location of experiment, it relates to the same feature - the relative content of chlorophyll measured by SPAD meter.

Authors’ Response: We have conducted the experiments in two different field conditions i.e, under drought stress and iron deficiency (Vijayanagar, Karnataka State, India). SPAD chlorophyll meter reading (SCMR) was recorded in both the experiments so that we can clearly correlate effect of specific stress on SCMR. Accordingly, the SPAD reading under drought stress experiment was denoted as SCMR-drought while it was denoted as SCMR-IDC in iron deficiency experiment. Now it is changed as SCMR-ID as per your suggestions.

REQUIRED / SUGGESTED MINOR CORRECTIONS:

  • I suggest you consider preparing a list of abbreviations

Authors’ Response: Thank you for this important suggestion. We have prepared a list of abbreviations and provided it at the last of a manuscript to avoid confusion.

Abbreviations

QTL

Quantitative trait locus

PVE

Phenotypic variation explained

GBS

Genotyping-by-sequencing

WGRS

Whole-genome re-sequencing

ddRAD-seq

Double digest restriction-site associated DNA

LOD

Logarithm of the odds

cM

 Centimorgan

Lob

Lateral organ boundaries

ABA

Abscisic acid

GH

Glycosyl hydrolase

LEA

Late embryogenesis abundant

bHLH

Basic helix-loop-helix

LRR

Leucine-rich repeat

NAC

NAM, ATAF and CUC

ROP

Rho of plants

MAS

Marker assisted selection

MYB

Myeloblastosis

  • The keywords should not contain the words contained in the title. Instead of repeating words, the keywords should contain:
    • the Latin name of the species
    • the usual name of the species, common synonymous to groundnut (peanut)
    • “abiotic stress” instead of “drought tolerance, iron deficiency chlorosis tolerance”
  • Formulation “map density”

Authors’ Response: Thank you for this important suggestion. We have updated the keywords as per your suggestions.

Keywords: abiotic stress; Arachis hypogaea; map density; SNP array; genetic map; genomics-assisted breeding; peanut

  • Abstract; line marked as line 50: parameter “2.2 cM/locus” is not an “average marker density” or “map density”. You should use the term “average distance between loci” (2.2 cM). Density can be referred to as the number of loci per (e.g.) 1 cM or 10 cM.

Authors’ Response: Thanks for bringing this to our notice. We have now replaced “average marker density” or “map density” word with “average distance between loci” in whole manuscript.

  • Table 2 “map distance”->”map length”, “map density” -> “average map distance”

Authors’ Response: Many thanks for this valuable suggestion. Now, we have replaced "map distance" word with “map length” and “map density” word with “average map distance”

  • “However, the majority of these studies failed to provide conclusive results due to a lack of optimum map density ( 56-347 loci).” -> However, the majority of these studies failed to provide conclusive results due to a lack of sufficiently extended genetic map (∼ 56-347 loci)

Authors’ Response: Thanks for this suggestion and we have replaced "However, the majority of these studies failed to provide conclusive results due to a lack of optimum map density (∼56-347 loci)" line with “However, the majority of these studies failed to provide conclusive results due to a lack of sufficiently extended genetic map (∼56-347 loci).

  • Section Phenotyping for drought and iron deficiency chlorosis (IDC) tolerance related traits, lines 9-14: “Later this population was also evaluated at Sadore (SD) (Niger) during 2009 and 2010 […]”.The previous sentence lists the years 2014-2015. 2009-10 is not “later” than 2014-15.

Authors’ Response: Thanks for bringing this to our notice. We have now corrected these lines in the text as given below.

“During 2014 and 2015, the population was phenotyped at ICRISAT-Patancheru (India) (PT) for days to 50% flowering, pod weight, 100 seed weight, haulm weight, and days to maturity. This population was also evaluated at Sadore (SD) (Niger) during 2009 and 2010, and at Bambey (BM) (Senegal) during 2009 for number of primary branches, plant height, SPAD chlorophyll meter reading (SCMR), Pod weight, haulm weight, harvest index, shelling percentage (SP), 100 kernels weight (100 KW), percentage of sound mature kernels (SMK %) under well-watered (WW) and water stress (WS) conditions [22].”

  • Section DNA extraction and genotyping with 58K SNPs ‘Axiom_Arachis’ array: It is enough to mention, that the DNA isolation was carried out in accordance with the manufacturer's instructions, or indicate only modifications (if any). The details in lines 5 – 15 are not needed.

Authors’ Response: Thanks for this important suggestion. We have reduced text here for DNA isolation and genotyping and kept minimum description for the benefit and clarity to the readers.

  • Among the references (e.g. in the discussion, next to items 46 and 47) there should be a work cited:

Hu, X.H., Zhang, S.Z., Miao, H.R. et al. High-Density Genetic Map Construction and Identification of QTLs Controlling Oleic and Linoleic Acid in Peanut using SLAF-seq and SSRs. Sci Rep 8, 5479 (2018). https://doi.org/10.1038/s41598-018-23873-7,

which concerns high density map of  the same species, refers to SLAF markers and QTLs of important traits.

Authors’ Response: We have cited above suggested reference (48) in the last line of first para in discussion section of the main text.

  • Discussion, line marked as 24: […] molecular breeding approach has been used to develop superior lines for oleic acid [38, 39] and foliar disease [40], such efforts have not been made in drought or IDC […] -> […] molecular breeding approach has been used to develop superior lines for oleic acid [38, 39] and foliar disease resistance [40], such efforts have not been undertaken for drought resistance or iron deficits resistance […]

Authors’ Response: Many thanks for this valuable suggestion. We have updated these lines in the main text and provided these as below:

In the case of groundnut, though molecular breeding approach has been used to develop superior lines for oleic acid [38, 39] and foliar disease resistance [40], such efforts have not been undertaken for drought or iron deficiency tolerance.

  • Discussion, line marked as 33,36: […] iron efficient genotypes […] -> […] iron uptake- efficient genotypes […]

Authors’ Response: Thanks for this suggestion and we have replaced "iron efficient genotypes " word with “iron uptake- efficient genotypes”.

  • Discussion, line marked as 85: Instead of "However [...]" -> "At the same time […]" would be more appropriate to start this sentence.

Authors’ Response: Thanks for this suggestion and we have replaced "However" word with “At the same time”.

  • Discussion, line marked as 96: "Surprisingly [...]".This requires an explanation of what the surprise is about. Surprisingly much?

Authors’ Response: We have replaced "Surprisingly" word with “Intriguingly”. We are provided this as below:

Intriguingly, in the present study, we identified total 551 E-QTLs for 12 drought tolerance related traits and 57 E-QTLs for 2 IDC tolerance related traits. Of these, 18 E-QTLs were identified as major E-QTLs for haulm weight, SCMR, and VCR.

  • Conclusions, line marked as 158: “The major effect QTLs identified for improving drought tolerance related traits namely haulm weight (qHW-A05.1)”. There is no QTL named “qHW-A05.1” in the main manuscript, only in Suplemmentary Table 2. Do you mean “qHW-A05.4”?

Authors’ Response: Many thanks for identifying this typo error. Mistakenly, qHW-A05.1 QTL is added in the text in place of qHW-A05.4 QTL. We have corrected it now.

MINOR NOTEs of EDITORIAL NATURE  

  • Section “Material and Methods” is numbered as “1” instead “2”

Authors’ Response: Thanks for this suggestion, however, as per the format of the Journal, the Introduction section comes as number 1 and then Materials & Methods as number 2.

  • Section 1. Main-effect QTLs for drought tolerance related traits: paragraph 2, line 2: Table 3 -> Table 4 (incorrect reference)

Authors’ Response:  Thanks for pointing out this error, we have now cited correct Table numbers in entire MS.

  • Table 3: all traits have full names, but TE. Expand abbreviation “TE

Authors’ Response: Many thanks for identifying this type of error, we have updated it now.

  • Figure 3: qPW-A03.7 -> qPW-A03.2

Authors’ Response: Thanks for pinpointing this typo error, we have replaced qPW-A03.7 with qPW-A03.2 in Figure 3.

  • Discussion, line marked as 128: catabolismrelated -> catabolism related

Authors’ Response: We have replaced "catabolismrelated" word with “catabolism related” from this sentence.

  • Discussion, line marked as 129: enzymeinvolved -> enzyme involved

Authors’ Response: We have replaced "enzymeinvolved" word with “enzyme involved” from this sentence.

  • Discussion, line marked as 140: andnicotian amine -> and nicotinamine

Authors’ Response: We have replaced " andnicotian amine" word with “and nicotinamine” from this sentence.

  • Discussion, line marked as 141-142 “Palmer et al. [69] reported under Fe deficiency in regulating the gene expression of nicotianaminesynthase genes” - grammar ?

Authors’ Response: We have reframed the line. We are provided this as below:

Palmer et al. [70] reported under Fe deficiency MYB10 and MYB72 act as a regulatory cascade to drive the gene expression of nicotianamine synthase genes (NAS2 and NAS4).

Reviewer 2 Report

The current article has a lot of information on subject, I feel like current version of manuscript should be considered for publication in Genes (MDPI).

But I have few comments,

  • The full forms of some of the candidate genes should be provided in result and abstract for example, MADS, Lob and so on

  • Elaborate information about best practices and summary only workflow need to be presented in methods

  • Reference should be given to some of the statements in results section 3.4. for implication example, [The 2- oxoglutarate/Fe (II)-dependent dioxygenases (Araip.IJN8L) genes encode family enzyme involved in various oxygenation/hydroxylation reactions, including the biosynthesis of Fe3+- chelating coumarines culetin, which is released into the rhizosphere for Fe uptake to maintain Fe homeostasis in plants (reference)].

  • In discussion or conclusion, future implication of current study should be written in detail.

Author Response

Response to the comments of the Reviewer#2

The current article has a lot of information on subject, I feel like current version of manuscript should be considered for publication in Genes (MDPI).

Authors’ Response: We thank the Reviewer for kind appreciation of our research work. We are grateful to him/her for providing valuable suggestions and important comments.

  • The full forms of some of the candidate genes should be provided in result and abstract for example, MADS, Lob and so on

Authors’ Response: Thanks for raising this point, we have updated abbreviation in text.

  • Elaborate information about best practices and summary only workflow need to be presented in methods

Authors’ Response: Thank you for this important suggestion. Axiom Analysis suite is a simple software to extract the genotyping data generated using SNP array. The best practice workflow and summary only are just once click pipelines for SNP calling in Axiom Analysis Suite. We have updated details of best practice and summary only workflow.

  • Reference should be given to some of the statements in results section 3.4. for implication example, [The 2- oxoglutarate/Fe (II)-dependent dioxygenases (Araip.IJN8L) genes encode family enzyme involved in various oxygenation/hydroxylation reactions, including the biosynthesis of Fe3+- chelating coumarines culetin, which is released into the rhizosphere for Fe uptake to maintain Fe homeostasis in plants (reference)].

Authors’ Response: The reference is cited in the main text at [67].

for implication example, [The 2- oxoglutarate/Fe (II)-dependent dioxygenases (Araip.IJN8L) genes encode family enzyme involved in various oxygenation/hydroxylation reactions, including the biosynthesis of Fe3+- chelating coumarines culetin, which is released into the rhizosphere for Fe uptake to maintain Fe homeostasis in plants [67].

  • In discussion or conclusion, future implication of current study should be written in detail.

Authors’ Response: Thank for this suggestion. We have provided more details on future implications of this study in Conclusion section.

Round 2

Reviewer 2 Report

Authors has addressed all the comments raised by me, I will recommend present form of article to publish in journal Genes MDPI.